# Using multi-modal neuroimaging to characterise social brain specialisation in infants

**Maheen Siddiqui[1]\*, Paola Pinti[1], Sabrina Brigadoi[2,3], Sarah Lloyd-Fox[4], Clare E Elwell[5], Mark H Johnson[4], Ilias Tachtsidis[5†], Emily JH Jones[1]\*†**

[1]Centre for Brain and Cognitive Development, Birkbeck, University of London, London, United Kingdom; [2]Department of Development and Social Psychology, University of Padova, Padova, Italy; [3]Department of Information Engineering, University of Padova, Padova, Italy; [4]Department of Psychology, University of Cambridge, Cambridge, United Kingdom; [5]Department of Medical Physics and Biomedical Engineering, University College London, London, United Kingdom

**\*For correspondence:**
m.siddiqui@bbk.ac.uk (MS);
e.jones@bbk.ac.uk (EJHJ)

†These authors contributed equally to this work

**Competing interest:** The authors declare that no competing interests exist.

**Abstract** The specialised regional functionality of the mature human cortex partly emerges through experience-dependent specialisation during early development. Our existing understanding of functional specialisation in the infant brain is based on evidence from unitary imaging modalities and has thus focused on isolated estimates of spatial or temporal selectivity of neural or haemo-dynamic activation, giving an incomplete picture. We speculate that functional specialisation will be underpinned by better coordinated haemodynamic and metabolic changes in a broadly orchestrated physiological response. To enable researchers to track this process through development, we develop new tools that allow the simultaneous measurement of coordinated neural activity (EEG), metabolic rate, and oxygenated blood supply (broadband near-infrared spectroscopy) in the awake infant. In 4- to 7-month-old infants, we use these new tools to show that social processing is accompanied by spatially and temporally specific increases in coupled activation in the temporal-parietal junction, a core hub region of the adult social brain. During non-social processing, coupled activation decreased in the same region, indicating specificity to social processing. Coupling was strongest with high-frequency brain activity (beta and gamma), consistent with the greater energetic requirements and more localised action of high-frequency brain activity. The development of simultaneous multimodal neural measures will enable future researchers to open new vistas in understanding functional specialisation of the brain.

## Editor's evaluation

This important study provides a state-of-the-art framework to explore the coupling of complementary cerebral measures (neural, hemodynamic and metabolic) during development by providing an interesting roadmap for multimodal neuroimaging in infants. The methodological contribution is compelling with an original setup for simultaneous EEG and NIRS recording and analyses. Results on the role of activation changes in the temporal-parietal junction on the development of social processing are convincing. This work will be of interest to a broad audience of scientists interested in multimodal neuroimaging and cognitive development.

## Introduction

The adult brain is highly specialised, with core networks coordinating to subserve complex behaviours. This specialised functioning emerges across development through a combination of genetically influenced brain architecture and experience-expectant learning processes (generalised neural development that occurs as a result of common experiences) and experience-dependent (variation in the environment contributing to individual differences in neural response) (*Johnson, 2001*). During early development, infants undergo significant neural, physiological, and socio-cognitive changes that are accompanied by large-scale changes in social communication and interaction. Currently, we have relatively few tools that allow us to comprehensively capture the emergence of functional specialisation in the infant social brain. Developing new approaches is critical for advancing our understanding of early brain physiology and cognitive function.

Identifying appropriate metrics to index functional specialisation in the infant brain should be informed by theoretical perspectives on how functional specialisation develops. Interactive specialisation is a theory of brain development that posits that functional specialisation emerges through competition between brain regions (*Johnson, 2011*). Thus, functional specialisation can be indexed as a smaller spatial extent of neural responses to a particular stimulus category and concomitant selectivity in responsive regions (*Jones et al., 2015*). Typically, the extent and selectivity of brain activation is measured through indirect indices of oxygenated blood flow (e.g. functional near-infrared spectroscopy [fNIRS] [*Bale et al., 2016*] or functional magnetic resonance imaging [fMRI] [*Attwell and Iadecola, 2002*]) or of coordinated neural activity (e.g. electroencephalography [EEG] [*Buzsáki, 2006*]). However, one mechanism that may contribute to competition between brain regions is the limited energetic resources available to the infant brain. The brain is an energetically costly organ, consuming 20–25% of the body's energy in adulthood while representing only 2% of the body's mass (*Raichle and Mintun, 2006*; *Sokoloff, 1999*). There are also substantial developmental changes in the brain's energy consumption; in the first year of life, up to 60% of available energy is used by the brain (*Steiner, 2019*). When brain regions become functionally active (e.g. during stimulus processing) neurons fire more rapidly, requiring greater supplies of adenosine triphosphate (ATP) (energy stores). Producing ATP requires oxygen, and this is supplied through a localised increase in oxygenated haemoglobin in the blood. Increases in oxygenated haemoglobin do not happen concurrently in all brain areas, and there are spatial dependencies between activated and deactivated regions in the adult brain (*Leech et al., 2014*). Energy supplies are important to synaptic plasticity, memory, and learning (*Vaynman et al., 2006*), and the mechanism through which energy supplies are coupled to activation (neurovascular coupling, see ) also develops through experience-dependent specialisation in the infant brain (*Kozberg and Hillman, 2016*). Thus, energy supply constraints may be one factor that contributes to the emergence of brain specialisation. If this is the case, detecting functional specialisation in infancy requires not only examining measures of neural activity and oxygenated haemoglobin, but also identifying whether particular regions show stronger coupling between neuronal demand and energetic supply.

As a first step, testing such frameworks requires the availability of methods that can measure the spatial extent and stimulus selectivity of neuroenergetics coupling in infancy. Previous studies have typically used single modalities sensitive to distinct aspects of brain function. For example, studies with fMRI indicate that core regions of the social brain (particular the fusiform face area) show increases in oxygenated haemoglobin delivery in response to faces by 4–9 months (*Kosakowski et al., 2022*). Further, fNIRS studies show that oxygenated haemoglobin delivery in response to naturalistic social videos in a broad region of temporal cortex emerges over the first hours of life (*Farroni et al., 2013*). Work with EEG indicates developmental increases in differentiated theta power responses to social versus non-social stimuli between 6 and 12 months (*Jones et al., 2015*). Thus, work with single modalities indicates development in functional specialisation across the first year of life.

Broadband near-infrared spectroscopy (bNIRS) is a new technique that uses a broad range of optical wavelengths which allows the measurement of the oxidation state of mitochondrial respiratory chain enzyme cytochrome *c* oxidase (CCO), thereby providing a direct measure of cellular energy metabolism (*Bale et al., 2016*). CCO is located in the inner mitochondrial membrane and serves as the terminal electron acceptor in the electron transport chain (ETC). It therefore accounts for 95% of cellular oxygen metabolism. In this way, bNIRS allows non-invasive measurement of cellular energy metabolism alongside haemodynamics/oxygenation in awake infants.

Work with single modalities has demonstrated that social selectivity in core regions of the adult 'social brain' can be robustly detected by 4–7 months of age (*Grossmann et al., 2010*; *Lloyd-Fox et al., 2012*; *Lloyd-Fox et al., 2013*; *Lloyd-Fox et al., 2018*). We recently showed the feasibility of using bNIRS in 4- to 7-month-old typically developing infants (*Siddiqui et al., 2021*) and demonstrated the presence of unique task-relevant, regionally specific functional networks where high levels of haemodynamic and metabolic coupling were observed. Here, we integrate this methodology with EEG to examine whether specific brain regions show coordinated energetic coupling and neural activity. We develop a novel analysis pipeline to identify localised coupling responses that are modulated by naturalistic social content. We aimed specifically to investigate the relationship between low- and high-frequency neural activity with haemodynamics and metabolism. For EEG, we expected an increase in neural activity in response to the social condition and a decrease in neural activity in response to the non-social condition. Based on previous work, this was expected to be strongest in the theta frequency band (*Jones et al., 2015*). Moreover, for the combined bNIRS-EEG analyses, we hypothesised differentiated haemodynamic/metabolic coupling with neural activity for the social and non-social stimulus conditions. We performed two types of statistical tests: (a) individual comparisons of the social and non-social conditions and (b) comparison of the social condition versus the non-social condition. The individual condition tests were performed to show the scale and spatial location/sensitivity of the coupling between haemodynamics/metabolism and neural activity for each condition. Meanwhile, the social versus non-social comparison was performed to show where there was a significant difference in the coupling between the two conditions. With comparison (a) we aimed to identify regions involved in the processing of social and non-social stimuli by identifying the regions where the coupling was significant. With comparison (b) we aimed to identify regions where coupling was significantly different between conditions. We predicted that for the individual comparison of the social condition, we would observe positive associations between bNIRS and EEG measures, that is a simultaneous increase in haemodynamics/metabolism and neural oscillatory activity in the beta and gamma frequency bands (based on previous combined EEG-fMRI studies *Scheeringa et al., 2011*; *Scheeringa et al., 2009*; *Goldman et al., 2002*; *Yuan et al., 2010*; *Niessing et al., 2005*; *Logothetis et al., 2001*; *Koch et al., 2009*) which would be localised to core social brain regions. We hypothesised that for the non-social condition, over the same brain regions, positive associations would be observed between bNIRS and EEG measures, but they would be a simultaneous decrease in haemodynamics/metabolism and oscillatory activity. We also expected simultaneous increases in haemodynamics/metabolism and oscillatory activity to be localised to the parietal brain region. These predictions are based on our previous work (*Siddiqui et al., 2021*) where we demonstrated that stronger coupling between haemodynamics and metabolism was observed in the temporo-parietal regions for the social condition and in parietal region for the non-social condition which is known to play an important role in object processing (*Wilcox et al., 2010*; *Dekker et al., 2011*). For the social versus the non-social contrast, we predicted that haemodynamic activity and metabolism would be coupled with neuronal oscillatory activity more strongly for the social stimuli in comparison to the non-social stimuli, with significant differences being observed in the temporo-parietal regions.

## Results

### Naturalistic social stimuli elicit expected increases in broadband EEG activity

Five-month-old infants (n=42) viewed naturalistic social and non-social stimuli (*Figure 1a*) while we concurrently measured EEG and broadband NIRS. Fourier transform of continuously recorded EEG data from 32 channels (n=35) in 1 s segments across the time course of stimulus presentation confirmed robust broadband increases in neural activity in response to social versus non-social stimuli (*Figure 1b*, replicating [*Jones et al., 2015*]).

### Haemodynamic and metabolic coupling and oscillatory activity spatially overlap

We used a method that we have previously validated to integrate haemodynamic and metabolic signals from the bNIRS data (n=25) to investigate the relationship between the two signals (*Siddiqui et al., 2021*; *Pinti et al., 2021*). Using this method, we obtained indices that indicated whether

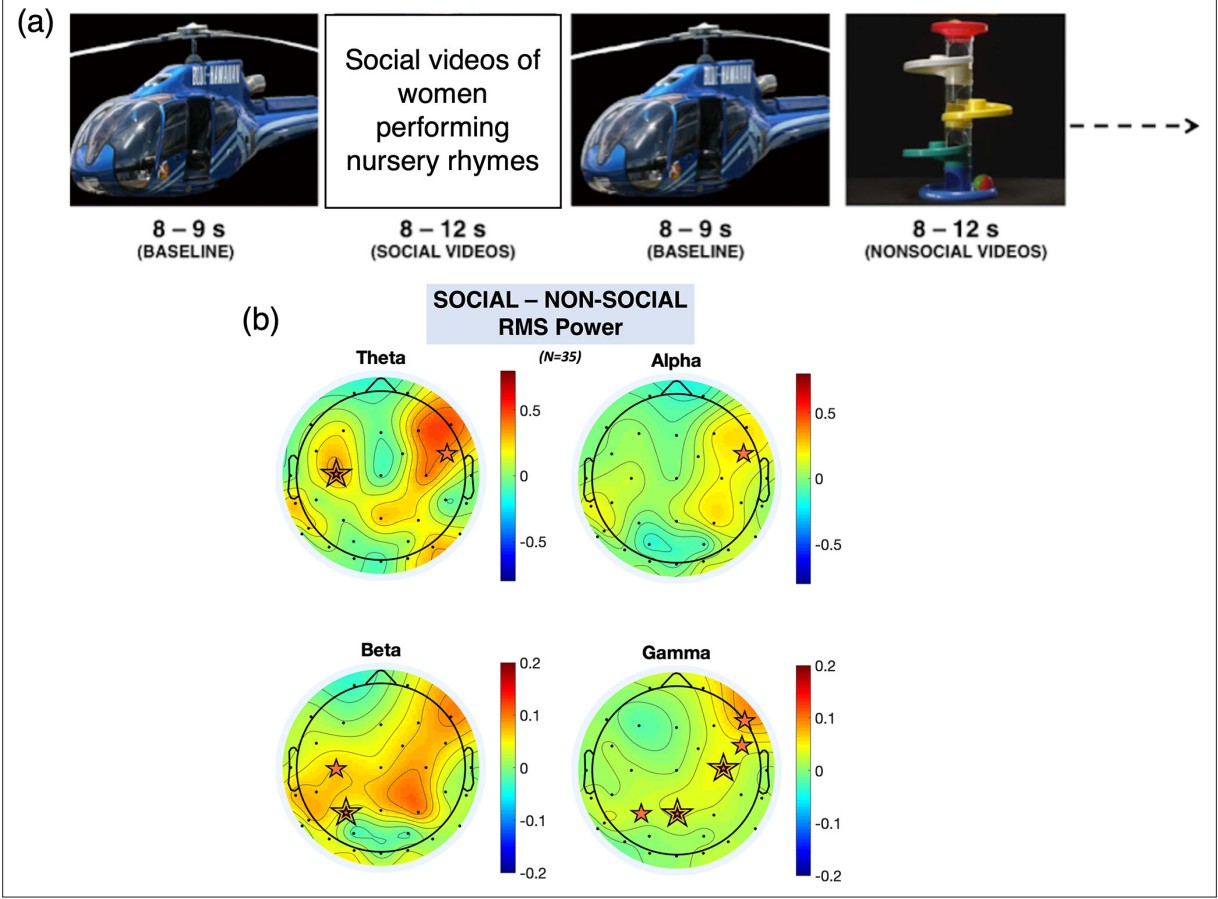

**Figure 1.** Order of stimulus presentation and EEG scalp topography. (**a**) Illustration of the paradigm. (**b**) Scalp topographies of the grand average root mean square (RMS) power for theta, alpha, beta, and gamma frequency bands (averaged across participants, averaged across the stimulus period) for the social minus non-social condition. The orange stars indicate statistically significant electroencephalography (EEG) electrodes where an increase in activity was observed (e.g. increase in response to the social condition compared to the non-social condition); a double line indicates significance after false discovery rate (FDR) correction.

specific brain regions either had a high level of coordinated coupling between haemodynamics and metabolism (i.e. coupled increases in metabolic function and oxygenated blood flow) or a mismatched coupling (i.e. an increase in metabolic function and a concurrent decrease in oxygenated blood flow). This revealed distinct locations sensitive to social (*Figure 2b*) and non-social (*Figure 2d*) processing; the topography of these locations is similar to the topography of differentiated broadband EEG activity (*Figure 2a and c*), particularly for haemodynamic and metabolic coupling at channels 12 and 14 and EEG theta band activity.

## Coupled signals highlight specialised activation in the temporal parietal junction

We then convolved the time course of the block-averaged within-hemisphere EEG time-series responses with an infant-specific haemodynamic response function (HRF) (n=14; *Figure 3*). A general linear model (GLM) approach was then used to identify false discovery rate (FDR)-corrected associations between all EEG locations and the bNIRS channels that showed significant coupling between the metabolic and haemodynamic response (*Figure 2b and d*). In line with the results shown in *Figure 2b* and *Figure 2d*, we expected the spatial coupling between bNIRS and EEG to differ for the social and non-social conditions. We predicted that for the social condition, we would observe coordinated increases in haemodynamic/metabolic activity (HbO₂ and oxCCO) and neural oscillatory activity (positive associations between bNIRS and EEG) in the beta and gamma frequency bands over the temporo-parietal region. Meanwhile, we expected that for the non-social condition, we would

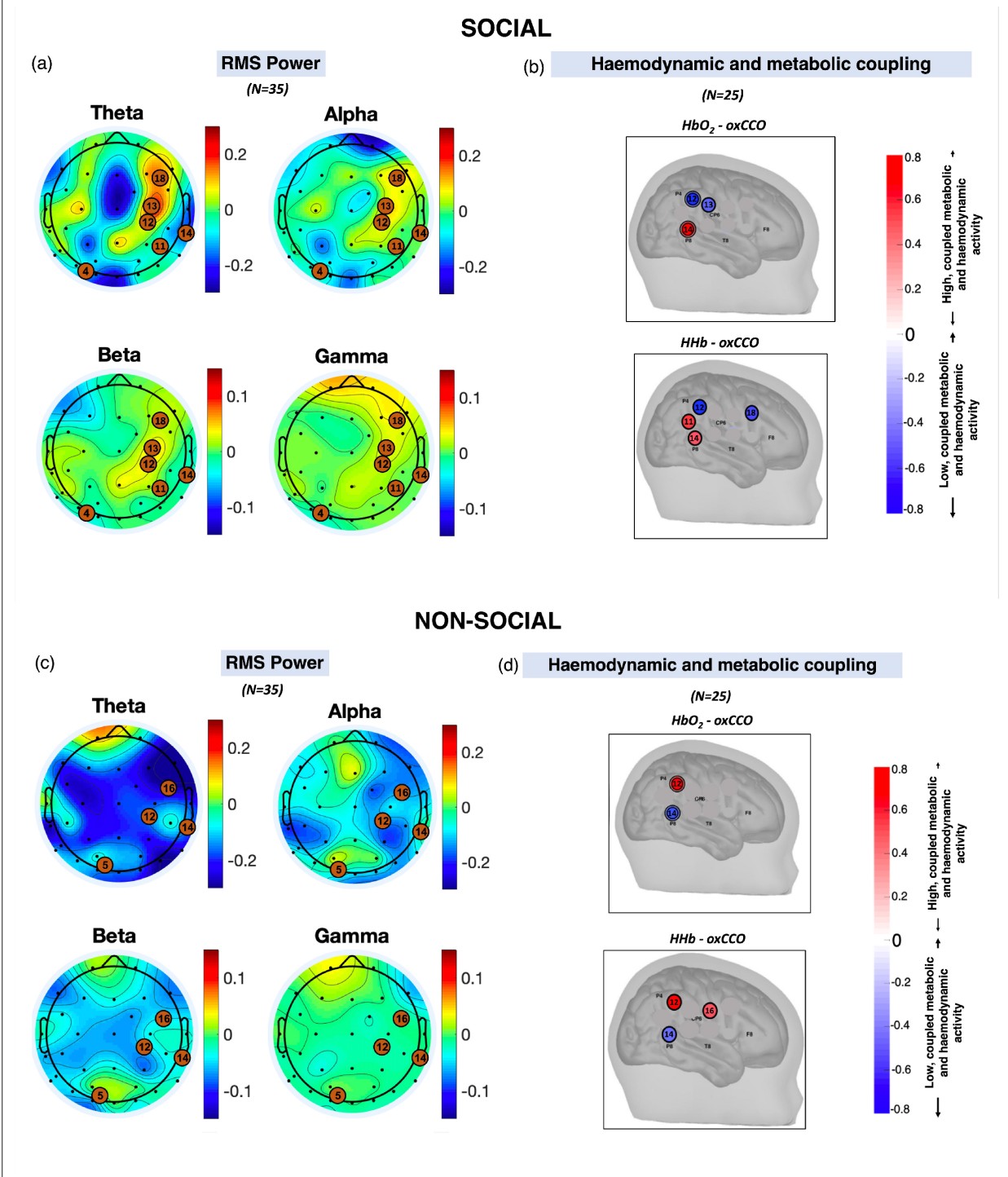

**Figure 2.** Scalp topographies of the grand average root mean square (RMS) power for theta, alpha, beta, and gamma frequency bands averaged across participants, averaged across the stimulus period for (**a**) social and (**c**) non-social conditions. The black dots show the locations of the electroencephalography (EEG) electrodes while the orange circles represent the broadband near-infrared spectroscopy (bNIRS) channels. Locations of high haemodynamic and metabolic coupling for (**b**) social and (**d**) non-social condition. (**b and d**) are reproduced from Figure 7 in *Siddiqui et al., 2021*.

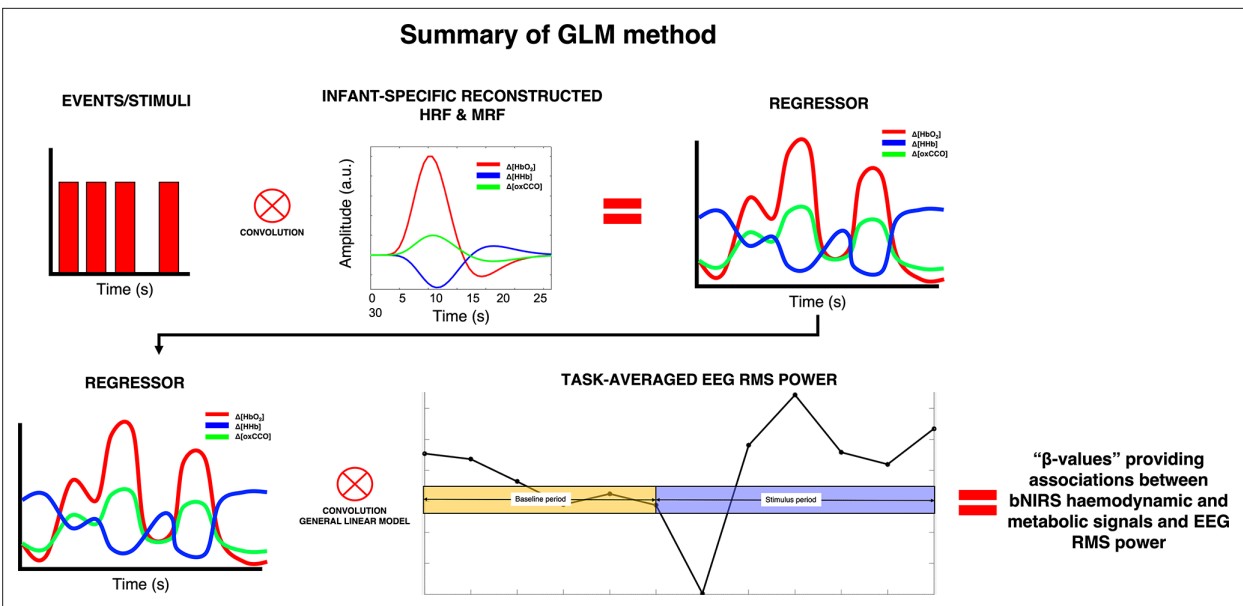

**Figure 3.** Summary of the procedure for obtaining the associations between broadband near-infrared spectroscopy (bNIRS) signals and electroencephalography (EEG) root mean square (RMS) power at each bNIRS combination, for each frequency band.

The online version of this article includes the following figure supplement(s) for figure 3:

**Figure supplement 1.** False discovery rate (FDR)-corrected significant connections between broadband near-infrared spectroscopy (bNIRS) channels (squares) and electroencephalography (EEG) electrodes (circles) for the (**i**) theta, (**ii**) alpha, (**iii**) beta, (**iv**) gamma, and (**v**) high gamma bands for the social condition (red colour bar) and the non-social condition (blue colour bar) for HbO$_2$, HHb, and oxCCO.

**Figure supplement 2.** False discovery rate (FDR)-corrected significant connections between broadband near-infrared spectroscopy (bNIRS) channels and electroencephalography (EEG) electrodes for the (**i**) theta, (**ii**) alpha, (**iii**) beta, (**iv**) gamma, and (**v**) high gamma bands for the social condition versus the non-social condition for HbO$_2$, HHb, and oxCCO.

observe coordinated decreases in haemodynamic/metabolic (HbO$_2$ and oxCCO) activity and neural oscillatory activity (also resulting in positive associations between bNIRS and EEG) over the temporo-parietal region and coordinated increases over the parietal region. We expected negative associations between HHb and oxCCO for both conditions. We predicted that the comparison of social versus non-social would show associations between bNIRS and EEG were stronger for the social condition.

*Figure 3—figure supplement 1* shows the individual statistical comparisons of the social (red colour scale) and non-social (blue colour scale) conditions. For both conditions, bNIRS-EEG coupling was consistently observed between bNIRS channel 14 and various EEG channels, which were positioned over the parietal and superior temporal sulcus-temporal parietal junction regions, respectively. For the social condition, a coupled increase in haemodynamic/metabolic activity and neural oscillatory activity was observed in the beta, gamma, and high-gamma frequency bands, which was primarily concentrated in the temporo-parietal region (e.g. bNIRS channel 14 and EEG electrodes Pz, PO4). A consistent pattern of coupling with neuronal activity was observed across chromophores particularly for the beta band. For the non-social condition, no coupling was observed between haemodynamics and neural activity (i.e. HbO$_2$ and HHb) for the low-frequency theta and alpha frequency bands. Meanwhile, a coupled increase in metabolic activity and neural activity was observed between bNIRS channel 14 and occipital and parietal EEG locations (O2, PO8, P10, P4 for the theta band and P10 for the alpha band). Moreover, in the high-frequency beta, gamma and high-gamma bands, coupling was observed primarily for HHb and oxCCO between bNIRS channel 14 and occipital, and parietal EEG locations (Oz, **O2, and PO8**). A consistent pattern of coupling was observed between HHb and oxCCO. Several long-range associations were also observed such as those in the beta frequency bands between bNIRS channels 12 and 13 and EEG locations TP8 and T8 respectively for the social condition for HbO$_2$ and between bNIRS channel 14 and EEG locations C2 and Cz for the non-social condition for HHb and oxCCO.

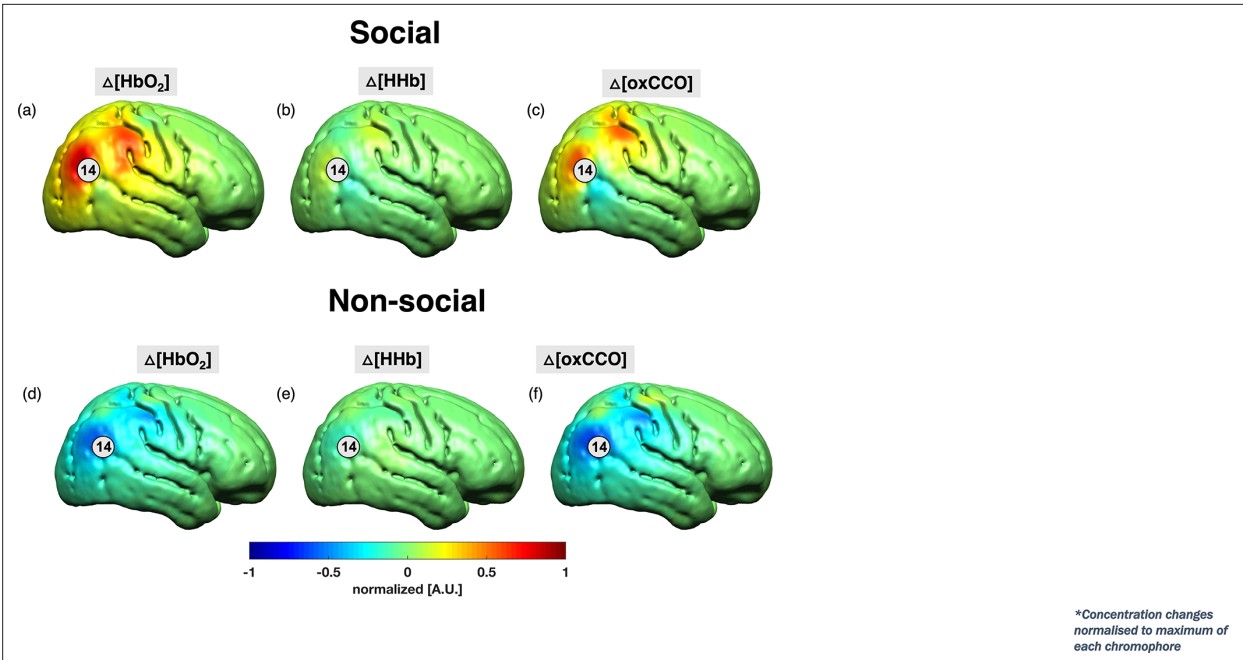

**Figure 4.** Grand average image reconstruction at 18 s post-stimulus onset for the social condition (a–c) and the non-social condition (d–f) at a single time point of 18 s post-stimulus onset. The concentration changes for $HbO_2$ and HHb were normalised to the maximum concentration change of $HbO_2$ while $\Delta oxCCO$ was normalised to its own maximum change in concentration. Channel 14 has been indicated.

*Figure 3—figure supplement 2* shows the statistical comparison of the social versus the non-social condition. Not many significant differences were observed between bNIRS and EEG associations for the two conditions. Significant differences were observed between bNIRS channel 14 and Pz (with stronger association for the social condition) in the gamma frequency band for $HbO_2$. Meanwhile, significant differences were observed between bNIRS channel 14 and O2 (with stronger association for the non-social condition) in the high-gamma band for oxCCO. This suggests differential coupling between haemodynamic/metabolic activity and neural activity for each condition.

Using image reconstruction on the bNIRS data, the spatial sensitivity of the bNIRS location that showed the clearest differences in coupling (channel 14) are shown in *Figure 4*. The method for image reconstruction has been described in detail in the Methods section. The results indicate that the bNIRS-EEG coupling was most consistent with the spatial extent changes in metabolic activity (CCO).

## Discussion

We develop a tool that enabled multimodal imaging analysis of coordinated neural activation, metabolic demand, and oxygenated haemoglobin delivery in the infant brain. As a proof of principle, we examined the relationship between these measures to identify regional selectivity to social versus non-social stimuli. To first demonstrate the scale and spatial sensitivity of the coupling between haemodynamic/metabolic activity and neuronal oscillatory activity, comparisons were performed individually for the social and non-social conditions. For this, we predicted a simultaneous increase in haemodynamics/metabolism and neural activity in the beta and gamma frequency band. We predicted that for the social condition this would be localised to the core social brain regions (temporo-parietal region) while for the non-social condition, we expected the coupling to be localised to parietal regions, known to be involved in object processing (*Wilcox et al., 2010*; *Dekker et al., 2011*). We additionally expected a simultaneous decrease in haemodynamic/metabolic activity and neural activity over the temporo-parietal region for the non-social condition, in accordance with our previous work (*Siddiqui et al., 2021*). Next, to demonstrate differential coupling for social and non-social stimuli, we performed a comparison of the social condition versus the non-social condition. For this, we hypothesised that in the beta and gamma frequency bands, there would be stronger coupling between haemodynamics/metabolism and neural activity for the social condition over the temporo-parietal region.

Confirming previous work, naturalistic social and non-social stimuli produce broad haemodynamic changes, with smaller spatial extent of locations with coupled haemodynamic and metabolic activity (*Siddiqui et al., 2021*). We also replicated previously observed greater EEG responses to social versus non-social stimuli (*Jones et al., 2015*). However, examining coupling between these two phenomena uncovered a precise pattern in which specific locations in the parietal and temporo-parietal regions showed differential coupling between bNIRS-EEG for social and non-social stimuli, particularly for the beta and gamma band frequency bands, as we predicted. We contend that this approach identifies a more localised regional area with selective coordination of neural, haemodynamic, and metabolic activity. The increased localisation observed in our coupling analysis may indicate our approach provides a more rigorous measure of functional specialisation. Widespread use of this technique will accelerate our understanding of both the typically and atypically developing brain. Unexpectedly, while most associations between haemodynamic/metabolic activity and oscillatory activity were localised, we observed several long-range connections between haemodynamic/metabolic and neural signals. It has been hypothesised that long-range functional connectivity patterns are vital for the organisation of human brain structure and function (*Wang et al., 2021*). The strongest coupling was observed between temporo-parietal bNIRS channel 14 with parietal EEG locations Pz and PO4 for the social condition (for beta and gamma frequency bands). Meanwhile, for the non-social condition, coupling was observed between temporo-parietal bNIRS channel 14 with occipital and parietal EEG locations Oz, O2, PO8, and P10 (for theta and beta frequency bands). While an overall consistent pattern of associations across chromophores and conditions was observed, some variability was also seen, particularly across frequency bands. This was expected and in line with previous EEG-fMRI studies that have demonstrated task-dependent variation in coupling between neural and haemodynamic activity across frequency bands (*Scheeringa et al., 2011*; *Scheeringa et al., 2009*; *Goldman et al., 2002*; *Yuan et al., 2010*; *Niessing et al., 2005*; *Logothetis et al., 2001*; *Koch et al., 2009*). For example, for resting-state simultaneous fMRI and EEG, stronger coupling between the BOLD response and neural activity has been observed for the alpha band (*Scheeringa et al., 2012*). Meanwhile, for cognitive tasks, stronger coupling has been observed in the gamma frequency band (*Kucewicz et al., 2014*). *Scheeringa et al., 2011*, investigated trial-by-trial coupling of EEG and BOLD activity and found that low- and high-frequency bands independently contribute to explaining BOLD variance. We therefore expected the frequency band showing the strongest coupling between bNIRS and EEG for each of the stimuli to vary. Further, while we did expect and observe significant overlap in associations between chromophores within each frequency band, some variability was seen. For example, for the social condition, no associations were observed in the low-frequency bands for any of the chromophores. Moreover, in the beta frequency bands, all chromophores displayed significant associations between bNIRS channel 14 and Pz for the social condition and both HHb and oxCCO displayed significant associations between bNIRS channel 14 and O2, PO8, and C2. Similarly, in the gamma frequency bands, both $HbO_2$ and oxCCO displayed significant associations between bNIRS channel 14 and PO4. The variability that was observed between chromophores was limited mostly to the non-social condition. For example, only oxCCO displayed significant associations between bNIRS and EEG for the low-frequency theta and alpha frequency bands. It is well known that various components involved in neurovascular coupling undergo development postnatally, see the review by *Harris et al., 2011*, for a full discussion. Briefly, there is an extensive structural change within cerebral microvasculature including growth, extension, and proliferation of new blood vessels (*Rowan and Maxwell, 1981*; *Norman and O'Kusky, 1986*). Further, studies have also demonstrated gradual development of vascular reactivity (i.e. change in vascular tone, vasoconstriction, and vasodilation) (*Scheeringa et al., 2011*; *Scheeringa et al., 2009*) which is necessary for the propagation of the NVC response (*Chen et al., 2014*). Lastly, pericytes and astrocytes which are key components of NVC are also known to undergo development in size, number, connectivity, and branching (*Kozberg and Hillman, 2016*; *Fujimoto, 1995*; *Seregi et al., 1987*; *Stichel et al., 1991*). From the metabolic perspective, infant positron emission tomography (PET) studies demonstrate regional-specific, progressive increase in the cerebral metabolic rate of oxygen consumption ($CMRO_2$) (*Chugani et al., 1987*) while others evidence a developmental maturational change in oxidative metabolism (*Kozberg and Hillman, 2016*). In adults, previous research has also suggested that oxygen consumption is more spatially localised in comparison to changes in cerebral blood flow (*Malonek and Grinvald, 1996*) and that oxCCO has distinct spatial distributions in the brain (*Phan et al., 2016*; *Wong-Riley et al., 1993*;

*Hevner et al., 1995*), indicating that energy metabolism may be more spatially specific. The spatial distribution of oxCCO in different brain regions currently remains unmapped in the developing infant brain, however. Therefore, taken together, given that during early development there are extensive changes in cerebral vasculature as well as the metabolic environment and potential variability in the spatial distribution of oxCCO, it is expected that there will be some variability observed in the associations between the haemodynamics and metabolism with neural activity. In our study, we observed more consistent oxCCO-EEG associations across frequency bands and stimuli with more localised (fewer long-range) associations. Further studies with a larger sample size and longitudinal follow-up can provide a clearer view on how NVC develops in infancy which will help explain some of the observed variability. Moreover, future studies with high-density bNIRS arrays will provide clarification on the spatial distribution of oxCCO in the infant brain.

EEG profiles observed in the present study are consistent with previous studies in identifying increased gamma band activity over temporal and parieto-occipital brain regions during face processing (*Uono et al., 2017*; *Ghuman et al., 2014*; *Bayer et al., 2018*; *Müller-Bardorff et al., 2018*; *Nguyen and Cunnington, 2014*; *Nguyen et al., 2014*; *Engell and McCarthy, 2011*; *Bossi et al., 2020*; *Anaki et al., 2007*; *Ishai et al., 2000*; *Pelphrey et al., 2005*; *Sato et al., 2014*; *Zion-Golumbic et al., 2008*; *Gao et al., 2013*). High-frequency neural firing is associated with localised processing (*von Stein and Sarnthein, 2000*) whilst lower-frequency activity is associated with larger-scale network changes and transfer of information across systems (*Canolty and Knight, 2010*). The increase in lower-frequency activity during social attention also observed here and in other work (*Jones et al., 2015*; *van der Velde et al., 2021*) may support larger-scale connectivity and communication of information through cross-frequency coupling (*Ghuman et al., 2014*). Our work further indicates that measures of metabolic load may provide important additional information in understanding localisation of brain function. Localised high-frequency activity exerts strong metabolic demand (*Smith et al., 2002*; *Kann, 2011*) and subsequent increases in oxygenated haemoglobin (*Logothetis et al., 2001*; *Niessing et al., 2005*; *Goense and Logothetis, 2008*). These increases in metabolic rate are supported by increased activity in the mitochondrial ETC, resulting in the changes in CCO we detected with broadband NIRS. Nitric oxide (which competes with oxygen to bind to CCO) and carbon dioxide (produced as a by-product in the ETC) are key signalling molecule in controlling neurovascular coupling and thus subsequent oxygen delivery (*Hosford and Gourine, 2019*; *Hosford et al., 2022*). Finally, reactive oxygen species produced by the ETC are a key signal in inducing synaptic plasticity (*Oswald et al., 2018*). Thus, our work is consistent with a model in which social attention induces localised high-frequency brain activity in the temporal parietal junction, which increases local metabolic rates, triggering synaptic plasticity and subsequent oxygen delivery to a broader region.

Our work particularly highlights the temporal-parietal junction is showing strong coupling and social selectivity. Previous studies measuring haemodynamic activity have identified early sensitivity of this region to social stimuli from at least 4 months (*Lloyd-Fox et al., 2017*), alongside a broader network of other regions. Here, we pinpoint this specific location as having coupled neuronal, metabolic, and haemodynamic activity that is modulated in opposite directions by complex social and non-social content. In the adult brain, the temporal-parietal junction has received considerable attention and there are several competing models of its function. It has been linked to mentalising (*Schurz et al., 2014*; *Schurz et al., 2017*) and reorienting attention to behaviourally relevant stimuli (*Corbetta and Shulman, 2002*); it can be viewed as a nexus area where the convergence of attention, language, memory, and social processing supports a social context for behaviour (*Carter and Huettel, 2013*) or as a region that is active when awareness of a prediction permits attentional control (*Wilterson et al., 2021*). Intriguingly, recent formulations within the predictive coding framework link the right temporal-parietal junction to a domain-general role in prediction, perhaps representing the precision of priors (*Masina et al., 2022*). Predictability has been linked to energy efficiency, with some computational models showing that energy limitations are the only requirement for driving the emergence of predictive coding (*Ali et al., 2021*). Increases in beta/gamma have also been linked to unexpected reward processing (*HajiHosseini et al., 2012*). Taken together, our results may indicate the early presence of priors for social interaction that are being actively updated (in contrast to the dynamic toys, which may already be more predictable).

The methods we developed could be broadly applied to study both neurotypical and atypical brain function. Assessing coupling over developmental time may reveal the mechanisms underpinning

neural specialisation and constrain theoretical frameworks seeking to explain specialisation in the adult brain. The mechanisms of neurovascular coupling remain unclear in the adult brain (*Hosford and Gourine, 2019*), and are developing in infancy (*Kozberg and Hillman, 2016*), and novel multimodal and non-invasive approaches to their identification could yield significant progress. Computational models could test the role of constraints in energy supply on developing localisation of function. Further, the region identified here also shows atypical haemodynamic responsiveness in infants with later symptoms of autism (*Lloyd-Fox et al., 2018*); since mitochondrial dysfunction has become an increasing focus in autism (*Siddiqui et al., 2016*) the possibility that atypical coupling may impact specialisation in autism is an important hypothesis to test. Further, our methods have applicability in determining the impacts of early brain injury. Recent work (*Bale et al., 2019*) measured both cerebral oxygenation and energy metabolism in neonates with brain injury (hypoxic-ischaemic encephalopathy) and demonstrated that the relationship between metabolism and oxygenation was able to predict injury severity. This therefore provided a clinical, non-invasive biomarker of neonatal brain injury. Indicating applicability across the lifespan (*Vezyroglou et al., 2022*) simultaneous measurements of cerebral oxygenation, metabolism, and neural activity in epilepsy revealed unique metabolic profiles for healthy brain regions in comparison to those with the regions of the epileptic focus. The work in epilepsy demonstrates the strength of combining measurements from multiple modalities to investigate brain states, particularly in clinical populations.

Our work has several limitations. We used naturalistic stimuli to maximise ecological validity; however, this reduces our ability to probe the function of the temporal-parietal junction across specific stimulus dimensions and this is an important target for future work. Limitations of current technology meant we recorded from the right hemisphere only and thus cannot determine the specificity of our findings to left temporal-parietal junction; engineering advances are required to produce whole-head bNIRS devices. Moreover, we only studied one age group of infants between 4 and 7 months; therefore, we could not investigate developmental change.

## Conclusion

Energy metabolism and neural activity are known to be tightly coupled in order to meet the high energetic demands of the brain, both during a task (*Jeong et al., 2006*; *Lundgaard et al., 2015*) and at rest (*Rocher et al., 2003*). It has been hypothesised that the level of correspondence between energy metabolism and neuronal activity may be an indicator for brain specialisation (*Jeong et al., 2006*; *Shokri-Kojori et al., 2019*; *Vaishnavi et al., 2010*). Here, we developed a system to simultaneously measure multichannel broadband NIRS with EEG in 4- to 7-month-old infants to investigate the neurovascular and neurometabolic coupling. We presented a novel study combining bNIRS and EEG and show stimulus-dependent coupling between haemodynamic, metabolic, and neural activity in the temporal-parietal junction. The results highlight the importance of investigating the energetic basis of brain functional specialisation and opens a new avenue of research which may show high utility for studying neurodevelopmental disorders and in clinical populations where these basic mechanisms are altered.

## Methods

### Participants

The study protocol was approved by the Birkbeck Ethics Committee, ethics approval number 161,747. Participants were forty-two 4- to 7-month-old infants (mean age: 179±16 days; 22 males and 20 females); parents provided written informed consent to participate in the study, for the publication of the research and additionally for the publication and use of any photographs taken during the study of the infant wearing the NIRS-EEG headgear. Inclusion criteria included term birth (37–40 weeks); exclusion criteria included known presence or family history of developmental disorders. The sample size was determined by performing a power analysis of existing data using G*Power.

### Experimental procedure

The experimental stimuli were designed using Psychtoolbox in Matlab (Mathworks, USA) and consisted of social and non-social videos. The social videos consisted of a variety of full-colour video clips of actors performing nursery rhymes such as 'pat-a-cake' and 'wheels on the bus'. The non-social videos

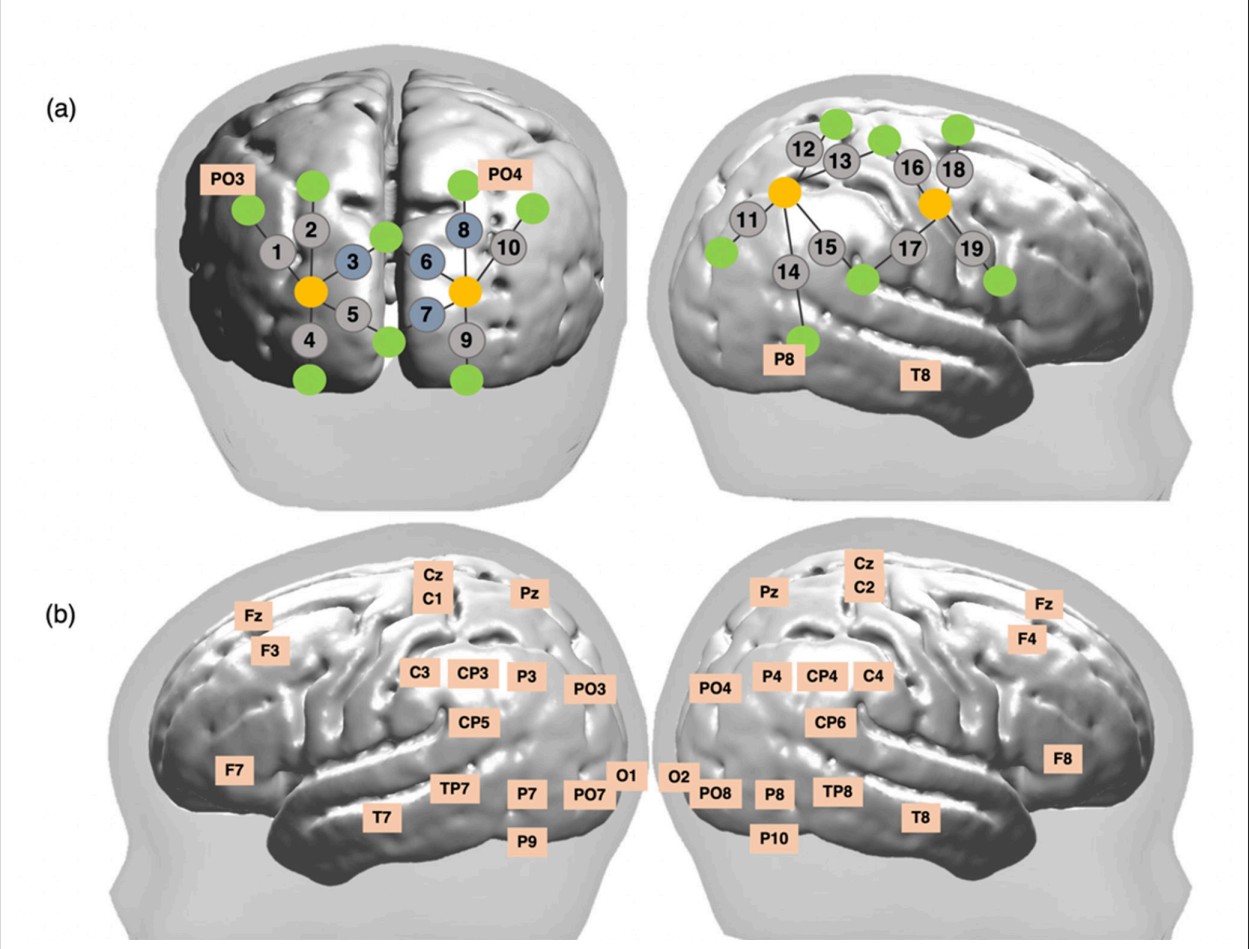

**Figure 5.** Schematic representation of broadband near-infrared spectroscopy (bNIRS) and electroencephalography (EEG) channel locations. (**a**) Locations of bNIRS channels (grey circles) over the occipital cortex and the right hemisphere and locations of the bNIRS sources (orange circles) and detectors (green circles) relative to EEG 10/20 locations. Channels shown in blue (3, 6, 8, and 10) were excluded from the analysis (**b**) Locations of the 32 EEG electrodes.

consisted of dynamic video clips of moving mechanical toys. The visual and auditory components of both social and non-social videos were matched. These videos have been used extensively in prior infant studies in both EEG (*Jones et al., 2015*) and NIRS studies (*Lloyd-Fox et al., 2009*; *Lloyd-Fox et al., 2014*). Both social and non-social experimental conditions were presented alternatingly for a varying duration between 8 and 12 s. The baseline condition consisted of static transport images, for example cars and helicopters, which were presented for a pseudorandom duration of 1–3 s each for a total of 8 s. Following the presentation of the baseline condition, a fixation cross in the shape of a ball or a flower appeared in the centre of the screen to draw the infant's attention back to the screen in case the infant had become bored during the baseline period. The following experimental condition was then presented once the infant's attention was on the fixation cross. *Figure 1a* depicts the order of stimulus presentation. All infants sat in their parent's lap at an approximate distance of 65 cm from a 35-in screen which was used to display the experimental stimuli. The study began with a minimum 10 s rest period to draw the infant's attention towards the screen during which the infant was presented with various shapes in the four corners of the screen. Following this, the baseline and experimental stimuli were presented alternatingly until the infant became bored or fussy.

## Data acquisition and array placement
bNIRS and EEG data was acquired simultaneously and the bNIRS optodes and EEG electrodes were positioned on the head using custom-built, 3D printed arrays which were embedded within a soft

neoprene cap (Neuroelectrics, Spain). *Figure 5a and b* shows the locations of bNIRS optodes and EEG electrodes on the head. *Figure 1b* shows the combined bNIRS-EEG headgear positioned on an infant. The array was designed to allow measurement from several cortical regions which included occipital, parietal, temporal, and central regions to allow investigation of neurovascular coupling in different cortical regions that are expected to be activated by dynamic stimuli.

## Broadband NIRS

Brain haemodynamic ($\Delta[HbO_2]$, $\Delta[HHb]$) and metabolic changes ($\Delta[oxCCO]$) were measured using an in-house broadband NIRS system developed at University College London (*Phan et al., 2016*). The bNIRS system consisted of two light sources that consisting of halogen light bulbs (Phillips) that emitted light in the near-infrared range (504–1068 nm). The light was directed to the infant's head through customised bifurcated optical fibres (Loptek, Germany), allowing each light source to split into two pairs of light sources. This formed a total of four light sources at the participant-end and each pair of light sources were controlled by a time multiplexing mechanism whereby one pair of light sources was on every 1.4 s. The system also consisted of 14 detector fibres at the participant-end which were connected to two spectrometers, 7 for each spectrometer (in-house developed lens spectrographs and PIXIS512f CCD cameras; Princeton Instruments). The configuration of 4 light sources and 14 detectors formed a total of 19 measurement channels. These were positioned over the occipital cortex and the right hemisphere as shown in *Figure 5a*. The source detector separation was 2.5 cm.

Data were analysed in Matlab (Mathworks, USA) using in-house scripts. First, for each participant, across all wavelengths, wavelet-based motion correction (*Molavi and Dumont, 2012*) was applied to the attenuation change signal to correct for motion artifacts. The tuning parameter $\alpha = 0.8$ was used. Following this, the UCLn algorithm (*Bale et al., 2016*) was used with a wavelength-dependent, age-appropriate fixed differential path-length factor value of 5.13 (*Duncan et al., 1995*). While the light sources emitted light between 504 and 1068 nm, the changes in concentration of $HbO_2$, HHb, and oxCCO were calculated using 120 wavelengths between 780 and 900 nm. A fourth-order bandpass Butterworth filter from 0.01 to 0.4 Hz was used to filter the data. For each infant, channels were assessed for signal quality and any channels with poor signal quality were rejected. Following this, the $HbO_2$, HHb, and oxCCO time-series were entered into a GLM to correlate bNIRS and EEG data.

For each infant, intensity counts (or photon counts) from each of the 14 detectors were used to assess the signal-to-noise ratio (SNR) at each channel and the channels with intensity counts lower than 2000 or higher than 40,000 were excluded (*Phan et al., 2016*). If an infant had more than 60% of channels excluded, they were excluded from the study. At the group level, five channels over the occipital cortex were excluded due to poor SNR in majority of infants (channel 3 excluded in 64% of infants, channel 6 excluded in 83% of infants, channel 7 excluded in 64% of infants, channel 8 excluded in 79% of infants) and one channel over the right hemisphere was excluded in 100% of infants due to a damaged optical fibre. The average number of blocks included at each channel was 6.

## EEG

EEG was used to measure neural activity simultaneously to haemodynamic and metabolic activity using the Enobio EEG system (Neuroelectrics, Spain) which is a wireless gel-based system. The system consisted of 32 electrodes, the locations of which are shown in *Figure 5b*. The sampling rate of the system was 500 Hz. The experimental protocol in Psychtoolbox sent event markers to both bNIRS and EEG systems using serial port communication which was then used to synchronise the bNIRS and EEG.

All data were analysed using the EEGlab Toolbox (Schwartz Centre for Computation Neuroscience, UC San Diego, USA) and in-house scripts in Matlab (Mathworks, USA). The raw EEG signal was bandpass filtered between 0.1 and 100 Hz and a notch filter (48–52 Hz) was applied to remove artefacts due to line noise. Following this, blocks of the data were created such that they consisted of the baseline period prior to the stimulus presentation and the entire following stimulus period. These blocks were then segmented into 1 s segments such that for both the baseline and the stimulus, each 8–12 s presentation of the baseline condition or the stimulus condition yielded 8–12 × 1 s segments. These 1 s segments consisted of 200 ms of the previous 1 s segment and 800 ms of the current segment and the 200 ms was used for baseline correction of each 1 s segment. This will be referred to as 'within-segment baseline correction' from here. Segments where the infants were not visually attending to the stimulus were removed. An average of 30 × 1 s segments were included per infant. Artefacts

were detected using automatic artifact detection in EEGlab and through manual identification. EEG segments were rejected if the signal amplitude exceeded 200 μV, or if electro-ocular, movement, or muscular artefacts occurred. Channels with noisy data were interpolated by an algorithm incorporated within EEGlab. Data were then re-referenced to the average reference.

Within each block (consisting of the baseline period and the stimulus period), each artefact-free 1 s segment was subjected to a power analysis to calculate the average root mean square (RMS) power for both low- and high-frequency bands – theta (3–6 Hz), alpha (8–12 Hz), beta (13–30 Hz), gamma (20–60 Hz), and high gamma (60–80 Hz), within each 1 s segment. This then yielded the average RMS power across the block (baseline period + following stimulus period). Baseline correction was performed by subtracting the average of the 2 s of the baseline period from the entire block. This will be referred to as the 'block baseline correction' from here on. RMS power was chosen as the metric to correlate bNIRS and EEG data as previous studies have demonstrated that task-related BOLD changes are best explained by RMS (*Kilner et al., 2005*; *Rosa et al., 2010*). The blocks were then averaged across trials to obtain an averaged RMS response per participant. A portion of the averaged RMS power was then entered into a GLM analysis described below – this consisted of 8 s of the stimulus period. *Figure 6—figure supplement 1* provides a visual depiction of how the RMS power was derived from the pre-processed EEG data. For each participant, the RMS power was also averaged across the stimulus period for statistical analysis of the EEG data. For each frequency band, statistical t-tests were performed on this averaged RMS power comparing the social condition versus the baseline (RMS power was averaged during the baseline period), the non-social condition versus the baseline, and social versus non-social. The FDR procedure using the Benjamin-Hochberg method (*Benjamini and Hochberg, 1995*) was performed to correct for multiple comparisons, across the 32 EEG channels.

## Data analysis
*Figure 6—figure supplement 2* outlines the data analysis pipelines for both bNIRS and EEG data, as well as the procedure for the combined bNIRS-EEG analysis.

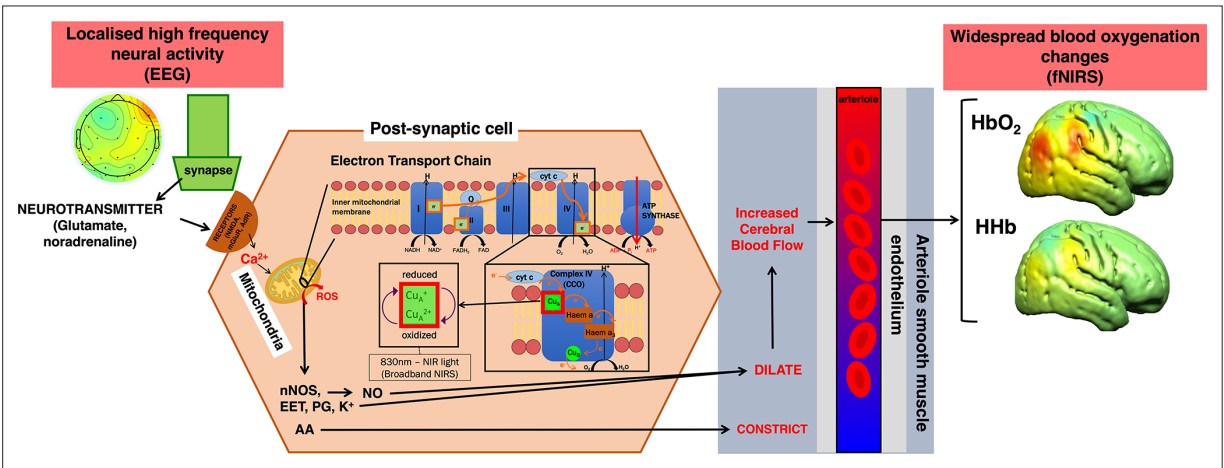

**Figure 6.** Simplified summary of the signalling pathways that mediate neurovascular coupling. High-frequency neural activity causes the release of neurotransmitters such as glutamate and noradrenaline which bind to either *N*-methyl-D-aspartate (NMDA) receptors in interneurons or metabotropic glutamate receptors (mGluR) or adrenaline receptors in astrocytes. In both cases, this causes an influx of calcium ($Ca^{2+}$) which in turn leads to an increase in adenosine triphosphate (ATP) production through the mitochondrial electron transport chain (ETC). As a by-product, in interneurons, nitric oxide (NO) is produced in the interneurons which dilates arterioles to increase blood flow leading to increased oxygen delivery in surrounding brain regions. Alternatively, in astrocytes derivates of arachidonic acid (AA) include prostaglandins (PG) and epoxyeicosatrienoic acids (EET) which cause vasodilation. This figure has been adapted from *Harris et al., 2011*.

The online version of this article includes the following figure supplement(s) for figure 6:

**Figure supplement 1.** Procedure for deriving the electroencephalography (EEG) root mean square (RMS) power from the pre-processed EEG data.

**Figure supplement 2.** Flowchart for the data analysis pipelines for broadband near-infrared spectroscopy (bNIRS) (left), electroencephalography (EEG) (middle), and combined bNIRS-EEG (right).

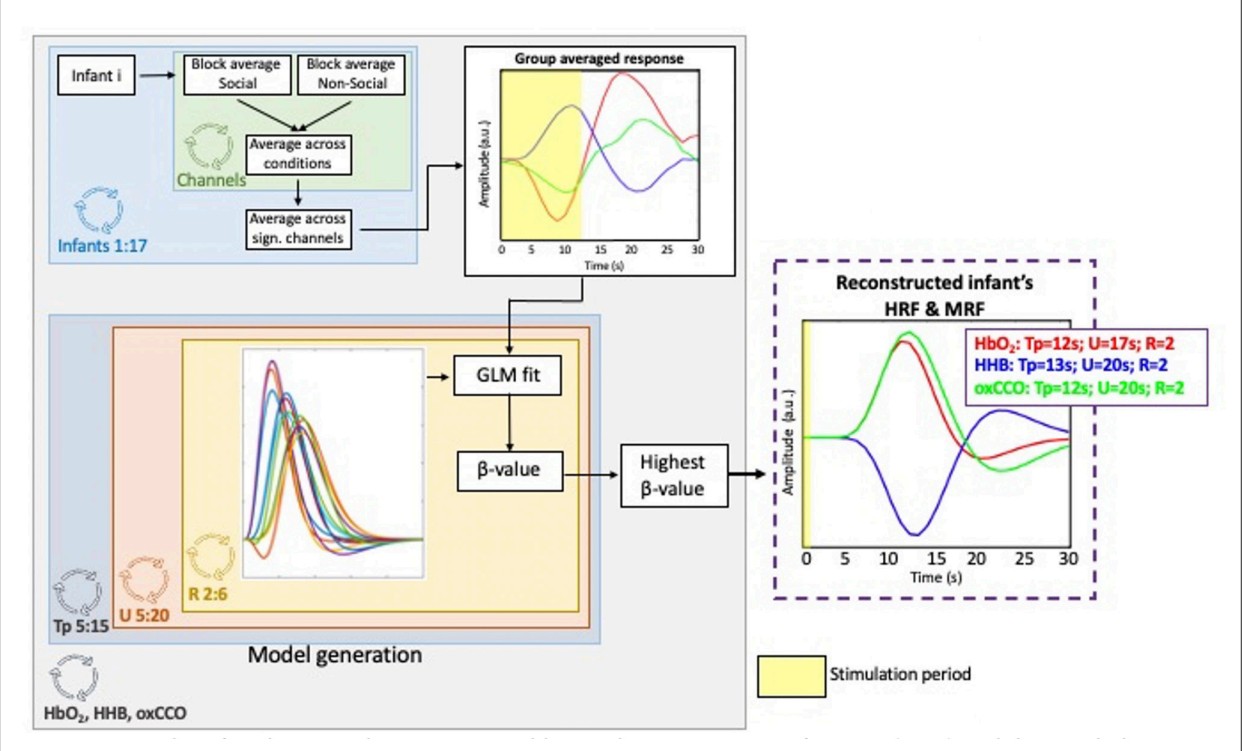

**Figure 7.** Procedure for obtaining the reconstructed haemodynamic response function (HRF) and the metabolic response function (MRF). The panel on the right shows the estimated HRF and MRF with the corresponding basis function parameters giving the best fit with the group-averaged HbO₂, HHb, oxCCO responses. The yellow shaded areas represent the stimulation periods.

## Combined NIRS-EEG analysis

A GLM (*Friston et al., 1994*) approach was employed to investigate the relationship between the bNIRS haemodynamic and metabolic data with the EEG neural data. *Figure 6* shows the physiological relationship between EEG neural activity and the bNIRS haemodynamic and metabolic signals. The canonical GLM typically uses a model of the expected haemodynamic response, that is the HRF, to predict the haemodynamic signal. However, given the differences in the haemodynamic response in adults and infants, the standard adult HRF model cannot be assumed for infant data. For example, infants display a delay in their haemodynamic responses (*Schroeter et al., 2004*; *Shimada and Hiraki, 2006*; *Minagawa-Kawai et al., 2011*). In addition, the analogous of the HRF is not established for the metabolic response (i.e. the metabolic response function [MRF]). Therefore, the first step of this analysis involved reconstructing the HRF for HbO₂ and HHb and the MRF for oxCCO before combing bNIRS and EEG data.

The reconstruction of the infant HRF and MRF started with block-averaging the HbO₂, HHb, and oxCCO signals for social and non-social conditions within each infant. Based on our previous study (*Siddiqui et al., 2021*), we selected only the channels that displayed statistically significant responses to the contrast task versus baseline. The single subjects block-averaged responses were averaged across the social and non-social conditions and then across the significant channels. The resulting block-averaged responses were then averaged across the group to obtain a 'grand average' HbO₂, HHb, and oxCCO response.

The grand average was then used in an iterative approach to estimate the HRF and MRF that best fit the HbO₂, HHb, and oxCCO responses. This involved fitting the grand-averaged signals with different HRF/MRF models starting from the canonical HRF made of two gamma functions and varying the following parameters: (a) delay of response, (b) delay of the undershoot, and (c) ratio of response to undershoot to identify the combination of parameters that best reconstructed the infant HRF/MRF for the social/non-social stimuli. The parameters were varied in increments of 1 s such that the delay of the response was varied from 5 to 15 s from the stimulus onset, the delay of the undershoot

was varied from 5 to 20 s, and the ratio of the response to the undershoot was varied from 2 to 6 s. All possible combinations of parameters were tested. The grand average responses were fitted with each HRF/MRF in GLM approach, and β-values were obtained for each combination of the HRF/MRF parameters. The β-values were entered into a statistical test and the parameter combinations that yielded the highest, statistically significant β-values (i.e. the model best fitting the data) were selected to reconstruct the infant HRF/MRF. This is approach is similar to those used previously to reconstruct the infant HRF (*Minagawa-Kawai et al., 2011*) and identified the best fit to be with a 2 s delay of response for $HbO_2$ and HHb and a 3 s delay of response for oxCCO in comparison to the adult HRF (i.e. 6 s). Moreover, the delay of the undershoot was 9 s earlier for all chromophores and the ratio of the response to the undershoot was 2 for $HbO_2$ and HHb and 3 for oxCCO, in comparison to 6 for the adult HRF. These correspond to the basis function representing the haemodynamic/metabolic response to an event of zero duration/impulse function. The new reconstructed HRF and MRF were then used for the GLM approach to correlate bNIRS and EEG data. The process for estimating the HRF and MRF has been depicted in *Figure 7*.

To constrain the analysis, we chose to investigate coupling of haemodynamic and metabolic with neural activity at specific channels. For this, we used the results from an analysis we described previously that combined bNIRS haemodynamic and metabolic signals (*Siddiqui et al., 2021*; *Pinti et al., 2021*). The results from this identified task-relevant cortical regions displayed high levels of haemodynamic and metabolic coupling. The bNIRS channels that displayed significant haemodynamic and metabolic coupling for social and non-social conditions were used here. All EEG channels were used as EEG is not as spatially specific as bNIRS. For each infant, for each chromophore, for each channel, and each EEG frequency band, the new infant HRF/MRF that was reconstructed in the previous step was convolved with the events to obtain the 'predicted' bNIRS signal. The 'predicted' bNIRS signal was then convolved with the EEG RMS power block (consisting only of the data from the stimulus period) at each frequency band to obtain the neural regressor for the bNIRS data, considering both social and non-social conditions together. The design matrix thus included the neural regressor reflecting the increased EEG activity to the social and non-social stimuli and used to fit the bNIRS data. This was performed for $HbO_2$, HHb, and oxCCO individually for all the channels. β-Values were estimated for each channel and t-tests against 0 were conducted to test whether there was a statistically significant association between bNIRS signals and EEG frequency bands. The FDR procedure using the Benjamin-Hochberg method (*Benjamini and Hochberg, 1995*) was performed to correct for multiple comparisons across EEG and bNIRS channels. The FDR-corrected significant t-values were plotted. This method has been used in numerous studies previously in correlating fMRI BOLD-EEG (*Scheeringa et al., 2009*). Only bNIRS channels that displayed significant (prior to FDR correction) haemodynamic and metabolic coupling were used for this analysis (as indicated in *Figure 2b and d*). For the social condition, channels 12, 13, and 14 for $HbO_2$, channels 11, 12, 14, and 18 for HHb, and channels 11, 12, 13, 14, and 18 for oxCCO displayed significant haemodynamic and metabolic coupling. Moreover, for the non-social condition, channels 12 and 14 for $HbO_2$, channels 12, 14, and 16 for HHb, and channels 12, 14, and 16 for oxCCO displayed significant coupling. For consistency, the channels selected for the bNIRS-EEG analysis were the same across chromophores and conditions. The final channels included in the analysis therefore were channels 11, 12, 13, 14, 16, and 18. For the integrated bNIRS-EEG analysis, six channel-wise t-tests were carried (one per included bNIRS channel, e.g. 6) for each EEG frequency band. Therefore, the FDR correction was applied across the six bNIRS channels for each of the hypotheses tested.

For the bNIRS analysis, data from 25 infants was included while for the EEG analysis, data from 35 infants were included. For the joint bNIRS-EEG analysis, only infants that had both valid bNIRS and EEG data for both social and non-social conditions were included and therefore 14 infants were included in this analysis.

## Image reconstruction

Image reconstruction was performed on the bNIRS data, at the individual subject level and then averaged across infants to produce a grand average that is shown in *Figure 4*. This was done to visually assess the similarities in the spatial distributions of the changes in $HbO_2$, HHb, oxCCO. For this analysis, three additional long-distance channels were created over the right hemisphere (in addition

to the 19 bNIRS channels) that had a source/detector separation of 4.3 cm to generate multiple and overlapping channels.

More precisely, the block-averaged attenuation changes at 13 discrete wavelengths (from 780 to 900 nm at 10 nm intervals) for each infant were selected from the bNIRS data. This was done to reduce the computational burden of the reconstruction (*Arifler et al., 2015*). A four-layer infant head model (consisting of the grey matter [GM], white matter, cerebrospinal fluid, and extra cerebral tissue) was built using averaged MRI data from a cohort of 12-month-old infants presented in *Shi et al., 2011*. The Betsurf segmentation procedure (*Jenkinson et al., 2005*) was then used to define an outer scalp boundary from the average head MRI template. The voxelised four-layer model was converted to a high-resolution tetrahedral mesh ($\sim$7.8$\times$10$^5$ nodes and $\sim$4.7$\times$10$^6$ elements) using the iso2mesh software (*Fang and Boas, 2009*). The same software was used to create the GM surface mesh ($\sim$5.8$\times$10$^4$ nodes and $\sim$1.2$\times$10$^5$ faces), used to visualise the reconstructed images.

The reconstruction of images of HbO$_2$, HHb, and $\Delta$oxCCO are described elsewhere (*Brigadoi et al., 2017*), using a multispectral approach (*Corlu et al., 2005*). Wavelength-specific Jacobians were computed with the Toast++ software (*Schweiger and Arridge, 2014*) on the tetrahedral head mesh and projected onto a 50 $\times$ 60 $\times$ 50 voxel regular grid for reconstruction, using an intermediate finer grid of 100 $\times$ 120 $\times$ 100 voxels to optimise the mapping between mesh and voxel space. Optical properties were assigned to each tissue type and for each wavelength by fitting all published values for these tissue types (*Bevilacqua et al., 1999*; *Strangman et al., 2002*; *Custo et al., 2006*). Diffuse boundary sources and detectors were simulated as a Gaussian profile with a 2 mm standard deviation, and Neumann boundary conditions were applied. The inverse problem was solved employing the LSQR method to solve the matrix equations resulting from the minimisation and using first-order Tikhonov regularisation, with the parameter covariance matrix containing the diagonal square matrices with the background concentration values of the three chromophores (23.7 for HbO$_2$, 16 for HHb, and 6 for $\Delta$oxCCO) (*Zhao et al., 2005*; *Bortfeld et al., 2007*) and the noise covariance matrix set as the identity matrix. The maximum number of iterations allowed to the LSQR method was set to 50, and with a tolerance of 10$^{-5}$. The regularisation hyperparameter $\lambda$ was set to 10$^{-2}$.

The reconstructed images, defined on the same regular grid of the Jacobian, were remapped to the tetrahedral head mesh and then projected to the GM surface mesh, by assigning a value to each node on the GM boundary surface that was equal to the mean value of all the tetrahedral mesh node values within a 3 mm radius. The concentration changes for HbO$_2$ and HHb were normalised to the maximum concentration change of HbO$_2$ while $\Delta$oxCCO was normalised to its own maximum change in concentration.

## Acknowledgements

MFS was funded by the BBSRC (BB/J014567/1), the Birkbeck Institutional Strategic Support Fund (ISSF), and the ESRC (ES/V012436/1). EJHJ was supported by the ESRC (ES/R009368/1). EJHJ, MHJ, and MFS were also supported by the AIMS-2-TRIALS programmes funded by the Innovative Medicines Initiative (IMI) Joint Undertaking Grant No. 777394. This Joint Undertaking receives support from the European Union's Horizon 2020 research and innovation programme, with in-kind contributions from the European Federation of Pharmaceutical Industries and Associations (EFPIA) companies and funding from Autism Speaks, Autistica, and SFARI. IT was supported by the Wellcome Trust (104580/Z/14/Z). SLF was supported by a UKRI Future Leaders Fellowship (MR/S018425/1) and SLF and CEE received support from the Bill and Melinda Gates Foundation (OPP1127625). MHJ and EJHJ received support from the UK Medical Research Council (MR/K021389/1 and MR/T003057/1). SB was supported by the Progetto STARS Grants 2017 (C96C18001930005) from the University of Padova. The work presented herein was conducted at the Centre for Brain and Cognitive Development, Birkbeck College, University of London. We are grateful to all the families who participated in this research and all the undergraduate students who assisted with data collection.

# Additional information

## Funding

| Funder | Grant reference number | Author |
|---|---|---|
| Biotechnology and Biological Sciences Research Council | [BB/J014567/1] | Maheen Siddiqui |
| Economic and Social Research Council | ES/V012436/1 | Maheen Siddiqui |
| Economic and Social Research Council | ES/R009368/1 | Maheen Siddiqui |
| Horizon 2020 Framework Programme | 777394 | Maheen Siddiqui Mark H Johnson Emily JH Jones |
| Wellcome Trust | 104580/Z/14/Z | Ilias Tachtsidis |
| UK Research and Innovation | MR/S018425/1 | Sarah Lloyd-Fox |
| Bill and Melinda Gates Foundation | OPP1127625 | Sarah Lloyd-Fox Clare E Elwell |
| Medical Research Council | MR/K021389/1 | Mark H Johnson Emily JH Jones |
| University of Padova | C96C18001930005 | Sabrina Brigadoi |
| Medical Research Council | MR/T003057/1 | Mark H Johnson Emily JH Jones |
| Wellcome Trust | 212979/Z/18/Z | Paola Pinti |

The funders had no role in study design, data collection and interpretation, or the decision to submit the work for publication. For the purpose of Open Access, the authors have applied a CC BY public copyright license to any Author Accepted Manuscript version arising from this submission.

## Author contributions

Maheen Siddiqui, Conceptualization, Data curation, Software, Formal analysis, Funding acquisition, Investigation, Visualization, Methodology, Writing – original draft, Project administration, Writing – review and editing; Paola Pinti, Software, Validation, Visualization, Methodology, Writing – original draft, Writing – review and editing; Sabrina Brigadoi, Formal analysis, Methodology, Writing – review and editing; Sarah Lloyd-Fox, Conceptualization, Supervision, Methodology, Writing – review and editing; Clare E Elwell, Conceptualization, Resources, Software, Supervision, Funding acquisition, Methodology; Mark H Johnson, Conceptualization, Resources, Supervision, Funding acquisition, Project administration, Writing – review and editing; Ilias Tachtsidis, Conceptualization, Resources, Software, Supervision, Funding acquisition, Methodology, Project administration, Writing – review and editing; Emily JH Jones, Conceptualization, Resources, Supervision, Funding acquisition, Validation, Visualization, Methodology, Writing – original draft, Project administration, Writing – review and editing

## Author ORCIDs

Maheen Siddiqui (ID) http://orcid.org/0000-0003-2037-6964
Sabrina Brigadoi (ID) http://orcid.org/0000-0003-3032-7381
Emily JH Jones (ID) https://orcid.org/0000-0001-5747-9540

## Ethics

Human subjects: The study protocol was approved by the Birkbeck Ethics Committee, ethics approval number 161747. Participants were forty-two 4-to-7-month-old infants (mean age: 179± 16 days; 22 males and 20 females); parents provided written informed consent to participate in the study, for the publication of the research and additionally for the publication and use of any photographs taken during the study of the infant wearing the NIRS-EEG headgear.

Decision letter and Author response
Decision letter https://doi.org/10.7554/eLife.84122.sa1
Author response https://doi.org/10.7554/eLife.84122.sa2

## Additional files

### Supplementary files
• MDAR checklist

### Data availability

The data contains human subject data from minors and guardians provided informed consent to having data shared only with researchers involved in the project, in anonymised form. A Patient and Public Involvement (PPI) initiative at the Centre for Brain and Cognitive Development aimed to actively work in partnership with parents and guardians participating in research studies to help design and manage future research. A comprehensive public survey was conducted as part of this initiative which aimed to evaluate parent attitudes to data sharing in developmental science. This survey revealed that majority of parents do not want their data to be shared openly but are open to the data being shared with other researchers related to the project. Therefore, in order to adhere to participant preference/ choice, a curated data sharing approach must be followed wherein the data can only be made available upon reasonable request through a formal data sharing and project affiliation agreement. The researcher will have to contact MFS via email on m.siddiqui@bbk.ac.uk and complete a project affiliation form providing their study aims, a detailed study proposal, plan for the analysis protocol, ethics, and plans for data storage and protection. Successful proposals will have aims aligned with the aims of the original study. Raw NIRS data, EEG data and integrated NIRS-EEG data can be made available in anonymised form. ID numbers linking the NIRS and EEG data, however, cannot be provided as parents/guardians have consented only to data being shared in anonymised form. All code used to analyse the NIRS data and the integration of the NIRS and EEG data is available on GitHub (https://github.com/maheensiddiqui91/NIRS-EEG, copy archived at *Siddiqui, 2023*). EEG data was processed using EEGlab which is a publicly available toolbox.

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
