## [Editor Report]

This important study provides a state-of-the-art framework to explore the coupling of complementary cerebral measures (neural, hemodynamic and metabolic) during development by providing an interesting roadmap for multimodal neuroimaging in infants. The methodological contribution is compelling with an original setup for simultaneous EEG and NIRS recording and analyses. Results on the role of activation changes in the temporal-parietal junction on the development of social processing are convincing. This work will be of interest to a broad audience of scientists interested in multimodal neuroimaging and cognitive development.

---

## [Decision Letter]

**Decision letter after peer review:**

Thank you for submitting your article "Mapping human social brain specialisation beyond the neuron using multimodal imaging in human infants" for consideration by *eLife*. Your article has been reviewed by 3 peer reviewers, and the evaluation has been overseen by a Reviewing Editor and Chris Baker as the Senior Editor. The following individual involved in the review of your submission has agreed to reveal their identity: Claire Kabdebon (Reviewer #3).

Essential revisions:

Although the reviewers acknowledged the important methodological contribution of your study, the following key aspects of the article were questioned and will need to be addressed in the revised manuscript, in addition to the other reviewers' comments (detailed below).

1) A discrepancy was perceived between the main claim of the article at the cognitive/developmental level (on brain specialization for social processing in infants) and the evidence provided by this work. It seems necessary to temper this presentation because the analyses and results are not as aligned with the aims and hypotheses as the reader might expect after reading the abstract and introduction. To gain clarity and impact of the manuscript, we advise you to partially reframe the article to better highlight the methodological contribution rather than emphasizing the fundamental side which is insufficiently supported by the results.

2) At the methodological level, avoiding shortcuts and general statements would be welcome. Better specification of your predictions in complex analyses such as the final general linear model would facilitate the interpretation of the results. The spatial comparison of source-localized NIRS measures and scalp-based EEG measures was also questioned.

3) Finally, during our discussions following the individual review process, Reviewer #2 suggested adding a critical discussion of the difference in hemodynamic response shape when signals are grand averaged vs. the HRF shape.

*Reviewer #1 (Recommendations for the authors):*

I feel the manuscript would be stronger if it focussed on evaluating the combination of methods. At the moment, it takes on too much and so ultimately disappoints. Theoretically, the introduction builds a case that this method might provide a special test of interactive specialization, but this is not revisited in the results or the discussion. And methodologically, there are large claims throughout (e.g., the title "beyond the neuron", line 73 "work with single modalities…does not yield insights", line 217 "a critical nexus in understanding localisation", line 254 "novel multimodal and non-invasive", line 268 "demonstrates the strength… particularly in clinical populations"). These are non-substantiated and distract from the genuine discoveries in the paper. Highly relevant literature is also omitted. For example, there is a whole community using EEG and fMRI, either sequentially or simultaneously, that has produced what I expect are hundreds of papers. What lessons have been learned from this? A summary should be given in the introduction and the work-related back in the discussion.

The authors perform source localisation to map the scalp measures to brain sources for the NIRS. It would be interesting to give some estimate of the spatial resolution of this method. But no such localisation is attempted for EEG. It was probably omitted because there are too few EEG channels for it to be meaningful. But, it is then just not possible to spatially compare scalp-based EEG measures, which may be far from the neural source to a neural measure, of localised NIRS. I feel that plotting the scalp electrodes on the brain, along with the NIRS data, is misleading.

It is difficult to read a paper where the methods are so important, with the methods at the end.

*Reviewer #2 (Recommendations for the authors):*

Please check two issues with Graphic links in the text: lines 374 and 561 "Error! Reference source not found."

*Reviewer #3 (Recommendations for the authors):*

Here are some additional remarks:

– In order to address the specialization process, and if the sample size and distribution allow, it might be interesting to explore age-related effects over the reported brain regions.

– About the organization of the methods section, for clarification, I would suggest following the structure of the Results section.

– Regarding the EEG analysis, some aspects of the analysis were unclear to me. First, about data segmentation: it is mentioned that data is epoched in 1s segments, consisting of 200ms of the previous segment and 800ms of the current segment: it is thus my understanding that segments are overlapping. However, it doesn't fit with the schematic representation of the procedure in figure 6b. This procedure should be clarified.

Second, it is also mentioned that the first 200ms of the segment was used as a baseline. What is the purpose of this baseline correction, since a second baseline correction will be performed in the frequency domain? Is it only for artifact detection purposes?

Third, regarding the definition of the frequency bands, the β and γ bands are overlapping (Β from 13 to 30Hz and Γ from 20 to 60Hz), which means that the two measures are not independent. How do the authors ensure that the reported effects in the two frequency bands are not driven by this shared activity?

– About the hemodynamic to metabolic coupling assessment, it is mentioned that the coupling of hemodynamic and metabolic activity was restricted to specific channels, based on previous studies. Which are these specific channels?

– Regarding the HRF and MRF reconstruction, the schematic representation of the procedure in figure 7 was confusing to me. The authors performed block averages of the HbO, HbR, and COO signals. However, it is unclear to me whether the blocks consist of [baseline stimulation] or [stimulation baseline]: what does the 0-time point correspond to in these figures? It seems like on the group average response on the top it represents the onset of the baseline, and on the reconstructed infant response functions it represents the onset of the stimulation. This is only a detail of the representation, but it was confusing for me.

Besides, do the depicted response functions represent the actual functions obtained? It would be informative to provide the reader with this information.

– Regarding the combined NIRS-EEG analysis, it was unclear to me whether the EEG measures were used to model the bNIRS data on a second-by-second basis (i.e. the time-resolved rms power signal is convolved with the response function), or on a block by block basis (i.e. average of the rms power signal for each block), or whether an average across each of the two conditions was used (i.e. average rms power across all blocks of each condition). Besides, L463-465 it is mentioned that the RMS power is entered in the GLM with 2s baseline + 8s stim, but L545, it is then mentioned that only the data for the stimulus period is included. Please clarify this point.

– Regarding multiple comparisons, the proposed analysis examining relations between EEG activity in 4 frequency bands to each of the three chromophores implies a high number of comparisons. The authors indicate that they used FDR corrections: across which dimensions were these corrections performed? Is it EEG x bNIRS sensors x chromophores x frequency bands? It would be worth mentioning for each of the analyses performed.

– Please mention the average amount of data used to perform each of the analyses across infants (i.e. how many blocks of stimulation are included).

– I found that some of the information included in the figures was sometimes too small or very hard to read. I would suggest re-designing some of these figures to be self-explanatory.

– Figure 2-f: it is unclear what this graph corresponds to, and how it was obtained. Does it correspond to a specific condition social or non-social? What is the added value of this graph in the framework of this study?

– There are some reference errors (L561 and 374).

[Editors’ note: further revisions were suggested prior to acceptance, as described below.]

Thank you for resubmitting your work entitled "Using multi-modal neuroimaging to characterise social brain specialisation in infants" for further consideration by *eLife*. Your revised article has been evaluated by Floris de Lange (Senior Editor) and a Reviewing Editor.

The manuscript has been improved but there are some remaining issues that need to be addressed, as outlined below.

*Reviewer #3 (Recommendations for the authors):*

In general, the authors produced a stronger manuscript that addresses most of the concerns that were raised. In particular, the authors improved the overall clarity of their work both conceptually and methodologically, and they provide better data visualization. Besides, the scope and contribution of the paper is now communicated more clearly.

Yet, a few aspects of the manuscript remain a bit obscure to me and I believe they should be further improved or at least addressed.

First, after reading the paper, the pattern of results obtained from the combined EEG – bNIRS analysis does not seem as clear as I would have expected from the authors predictions. In figures 3-1 and 3-2, although some bNIRS channels (11 and 14) are overall more consistently reported than others as significant, as emphasized in the results and discussion, there is still quite some variability in the pattern of bNIRS to EEG channels coupling across chromophores (and frequency bands and conditions). How do the authors explain this variability? Shouldn't the different chromophore measurements reflect a similar phenomenon, and thus display similar coupling patterns with EEG channels? Is this the expected level of variability for this kind of dataset, or for this age range, population size etc….? It would be really informative if the authors could elaborate more on the reasons why we observe this variability in the discussion, and if possible provide recommendations on how to reduce it in further studies.

Second, about the authors' predictions in the introduction, they mention that they expect to observe "coordinated increases in hemodynamics/metabolism and neural oscillatory activity" in the social condition, and "coordinated decreases in hemodynamics/metabolism and neural oscillatory activity" in the non-social condition. This wording suggests that this is a hypothesis that the authors will be able to test. However, the authors do not test this hypothesis further in the manuscript and it seems to me that the directionality of coordinated activity cannot be directly assessed using this GLM approach. I would therefore suggest to slightly reframe the phrasing of this paragraph to avoid that confusion.

---

## [Author Response]

Essential revisions:1) A discrepancy was perceived between the main claim of the article at the cognitive/developmental level (on brain specialization for social processing in infants) and the evidence provided by this work. It seems necessary to temper this presentation because the analyses and results are not as aligned with the aims and hypotheses as the reader might expect after reading the abstract and introduction. To gain clarity and impact of the manuscript, we advise you to partially reframe the article to better highlight the methodological contribution rather than emphasizing the fundamental side which is insufficiently supported by the results.

We would like to thank the editors for their comments. Pertaining to the first point, the introduction and Discussion sections of the article have been reframed to emphasize the methodological impact. We have also changed the title to reflect an increased methodological focus. We have retained the theoretical background that motivates the development of the measures but made it clearer that the primary contribution of this article is in developing these methods.

2) At the methodological level, avoiding shortcuts and general statements would be welcome. Better specification of your predictions in complex analyses such as the final general linear model would facilitate the interpretation of the results. The spatial comparison of source-localized NIRS measures and scalp-based EEG measures was also questioned.

We have removed any general statements and have included hypotheses to facilitate interpretation of results. Further, to clarify, we did not perform any source-localisation on the NIRS data and therefore we did not compare scalp-based EEG and source-localised NIRS. We have added text in the article to ensure this is clear.

3) Finally, during our discussions following the individual review process, Reviewer #2 suggested adding a critical discussion of the difference in hemodynamic response shape when signals are grand averaged vs. the HRF shape.

Regarding the last point, we have clarified both in the text (Line 1554-1555) and in Figure 7 that the estimated HRF and MRF correspond to the basis function to be used within the GLM (i.e. analogous to the canonical HRF typically used for fMRI and fNIRS GLM-based analyses and convolved with boxcar functions to generate regressors). These hence correspond to the hemodynamic/metabolic response to an event of 0 duration and differ from the group averaged responses that represent the responses averaged across the whole duration of the various task blocks. The HRF and MRF were estimated using the iterative procedure as these are not known for infants and the MRF nor for infants and adults, therefore providing a new method to optimize the analysis of NIRS-based data when the canonical HRF might not be appropriate.

Reviewer #1 (Recommendations for the authors):I feel the manuscript would be stronger if it focussed on evaluating the combination of methods. At the moment, it takes on too much and so ultimately disappoints. Theoretically, the introduction builds a case that this method might provide a special test of interactive specialization, but this is not revisited in the results or the discussion. And methodologically, there are large claims throughout (e.g., the title "beyond the neuron", line 73 "work with single modalities…does not yield insights", line 217 "a critical nexus in understanding localisation", line 254 "novel multimodal and non-invasive", line 268 "demonstrates the strength… particularly in clinical populations"). These are non-substantiated and distract from the genuine discoveries in the paper. Highly relevant literature is also omitted. For example, there is a whole community using EEG and fMRI, either sequentially or simultaneously, that has produced what I expect are hundreds of papers. What lessons have been learned from this? A summary should be given in the introduction and the work-related back in the discussion.

We thank the reviewer for their comment. We revised the introduction to increase the focus on the methodological advances represented by our work, though we think the broader theoretical context that motivates the use of these methods in general is still important to consider (because that provides the motivation for developing them). We have clarified that we are suggesting that integrating metabolic measurements and their coupling to other measures may provide a more sensitive way of measuring functional specialisation (the selective engagement of a particular brain region in a particular process) because energetic constraints may play a role in interactive specialisation, one theory of how functional specialisation arises. We are not arguing that our measures can measure interactive specialisation itself, and so we have tried to highlight this distinction throughout. We have further clarified that this is the context in which we present material on theoretical frameworks and returned to them in the discussion from this perspective. We have also clarified several places where we used unclear phrasing (e.g., on line 1148 we meant that the study we were discussing on epilepsy had implications for use in clinical populations, rather than our study directly; we have clarified this now). We have added material on simultaneous EEG and fMRI to the introduction and discussion.

The authors perform source localisation to map the scalp measures to brain sources for the NIRS. It would be interesting to give some estimate of the spatial resolution of this method. But no such localisation is attempted for EEG. It was probably omitted because there are too few EEG channels for it to be meaningful. But, it is then just not possible to spatially compare scalp-based EEG measures, which may be far from the neural source to a neural measure, of localised NIRS. I feel that plotting the scalp electrodes on the brain, along with the NIRS data, is misleading.

We thank the reviewer for their comment. To clarify, we did not perform source localisation on the NIRS data for our analyses examining associations with EEG. We did perform image reconstruction (as seen in Figure 4) with the bNIRS data, but this was to address a separate goal. The HbO_2_, HHb and oxCCO time-series (i.e. for each channel) were used as part of the combined bNIRS – EEG analysis, not the reconstructed time-series (i.e. for each node of the mesh). We have added text on lines 1623 – 1624 to make it clear that image reconstruction was a separate analysis that was performed with the main goal of visualizing how the changes in HbO_2_, HHb, oxCCO were spatially related to each other. Moreover, we have changed Figure 6c to indicate at which step of the analysis image reconstruction was performed. This should also clarify that the data from the reconstruction was not used in any other subsequent analyses.

It is difficult to read a paper where the methods are so important, with the methods at the end.

We have followed the *eLife* template for this.

Reviewer #2 (Recommendations for the authors):Please check two issues with Graphic links in the text: lines 374 and 561 "Error! Reference source not found."

This has been fixed.

Reviewer #3 (Recommendations for the authors):– In order to address the specialization process, and if the sample size and distribution allow, it might be interesting to explore age-related effects over the reported brain regions.

We thank the reviewer for their suggestion which is indeed interesting and will be considered for future studies. Due to sample size limitations, we do not believe that age-related effects can be explored within the current dataset. However, our methods provide protocols for investigators to address this question in future studies.

– About the organization of the methods section, for clarification, I would suggest following the structure of the Results section.

We thank the reviewer for their suggestion. In its current form, the EEG results are presented first in the Results section, followed by the combined bNIRS – EEG results. The bNIRS results are not shown as they have already been published in a previous publication. Moreover, in the methods section, the bNIRS methodology is detailed first, followed by the EEG and then the combined bNIRS – EEG. This follows the same structure as the Results section.

– Regarding the EEG analysis, some aspects of the analysis were unclear to me. First, about data segmentation: it is mentioned that data is epoched in 1s segments, consisting of 200ms of the previous segment and 800ms of the current segment: it is thus my understanding that segments are overlapping. However, it doesn't fit with the schematic representation of the procedure in figure 6b. This procedure should be clarified.

The reviewer is correct in their understanding that the segments are overlapping. This has not been shown in Figure 6b as it would make the figure unnecessarily cluttered and difficult to understand. The point about overlapping segments has been emphasised in the caption of the figure.

Second, it is also mentioned that the first 200ms of the segment was used as a baseline. What is the purpose of this baseline correction, since a second baseline correction will be performed in the frequency domain? Is it only for artifact detection purposes?

It is standard procedure in EEG data analysis to perform baseline correction to get rid of drifts in the data and for the purposes of artifact detection.

Third, regarding the definition of the frequency bands, the β and γ bands are overlapping (Β from 13 to 30Hz and Γ from 20 to 60Hz), which means that the two measures are not independent. How do the authors ensure that the reported effects in the two frequency bands are not driven by this shared activity?

The frequency bands that are used are based on previous work by Saby and Marshall (2012) where infant frequency bands have been defined. These bands are widely adopted by EEG infant studies. In order to clear any doubts, we have now included a “high γ” frequency band (60 – 80 Hz) in the combined bNIRS – EEG analysis. The results shown in Figures 3b and 3c show consistency/similarity in the pattern of results in the “γ” and “high γ” frequency bands and no similarity between the “β” and “γ” frequency bands. We are therefore confident that the activity between the two bands is independent.

– About the hemodynamic to metabolic coupling assessment, it is mentioned that the coupling of hemodynamic and metabolic activity was restricted to specific channels, based on previous studies. Which are these specific channels?

This has been mentioned on line 1505 in the manuscript. To clarify, for each condition, for each chromophore, a different set of channels showed significant haemodynamic and metabolic coupling. To avoid further complexity, we decided to include all the channels that showed significant coupling for either of the conditions, for any of the chromophores. Therefore, considering all the significant channels across conditions and chromophores, the final channels included were 11, 12, 13, 14, 16 and 18.

– Regarding the HRF and MRF reconstruction, the schematic representation of the procedure in figure 7 was confusing to me. The authors performed block averages of the HbO, HbR, and COO signals. However, it is unclear to me whether the blocks consist of [baseline stimulation] or [stimulation baseline]: what does the 0-time point correspond to in these figures? It seems like on the group average response on the top it represents the onset of the baseline, and on the reconstructed infant response functions it represents the onset of the stimulation. This is only a detail of the representation, but it was confusing for me.Besides, do the depicted response functions represent the actual functions obtained? It would be informative to provide the reader with this information.

We thank the reviewer for the comment. The block in the group-averaged responses corresponds to the period [stimulation + baseline] as infants often show delayed responses (e.g. see Lloyd-Fox et al., 2017 using a similar task). The HRF and MRF on the right of the figure instead represent the basis function, i.e. the model of the response to an event of 0 duration. In addition, the figure shows the actual functions obtained. In order to clarify these aspects, we have included the yellow shaded areas to indicate the stimulation periods as well as mentioned that the shown HRF and MRF are estimated functions.

– Regarding the combined NIRS-EEG analysis, it was unclear to me whether the EEG measures were used to model the bNIRS data on a second-by-second basis (i.e. the time-resolved rms power signal is convolved with the response function), or on a block by block basis (i.e. average of the rms power signal for each block), or whether an average across each of the two conditions was used (i.e. average rms power across all blocks of each condition). Besides, L463-465 it is mentioned that the RMS power is entered in the GLM with 2s baseline + 8s stim, but L545, it is then mentioned that only the data for the stimulus period is included. Please clarify this point.

We thank the reviewer for their comment. To clarify, the average RMS power (which is referred to as the “block averaged EEG response” in the text) across all blocks of each condition was used for the convolution with reconstructed infant HRF/MRF. This is because unlike fNIRS/bNIRS data, several trials of EEG data are required to accurately detect activity. It was therefore more accurate to average all the blocks of each condition for each participant. This has already been mentioned on line 1350 – “The blocks were then averaged across trials to obtain an averaged RMS response per participant” Regarding the reviewer’s second comment, there was indeed a discrepancy, and we had erroneously stated that the RMS power block that was entered into the GLM consisted of 2s baseline + 8s stimulus. In fact, only the stimulus period was used for the GLM analysis. This error has been corrected now on line 1352.

– Regarding multiple comparisons, the proposed analysis examining relations between EEG activity in 4 frequency bands to each of the three chromophores implies a high number of comparisons. The authors indicate that they used FDR corrections: across which dimensions were these corrections performed? Is it EEG x bNIRS sensors x chromophores x frequency bands? It would be worth mentioning for each of the analyses performed.

For the EEG analysis, the FDR correction was performed within each frequency band, across EEG channels. For the combined bNIRS – EEG analysis, the FDR correction was performed across EEG channels x bNIRS channels. This has been added to line 669 of the manuscript.

– Please mention the average amount of data used to perform each of the analyses across infants (i.e. how many blocks of stimulation are included).

6 blocks are used per infant on average for the bNIRS data. For the EEG data, an average of 30 1-s segments were used. This has been mentioned on line 1307 and 1326.

– I found that some of the information included in the figures was sometimes too small or very hard to read. I would suggest re-designing some of these figures to be self-explanatory.

We have tried to re-design the figures as and where possible. For example, for Figure 3, where the combined bNIRS – EEG results are presented, we have put the social and non-social results on one figure with two colour scales for ease of comparison. We have also stated explicitly what each colour represents to make it easier to interpret. Due to the number of results (bNIRS channels, EEG electrodes, frequency band and chromophores) it is difficult to make the results figures very big.

– Figure 2-f: it is unclear what this graph corresponds to, and how it was obtained. Does it correspond to a specific condition social or non-social? What is the added value of this graph in the framework of this study?

We thank the reviewer for their comment and have removed Figures 2e and 2f as they did not add any value to the manuscript.

– There are some reference errors (L561 and 374).

These have been checked.

[Editors’ note: what follows is the authors’ response to the second round of review.]

The manuscript has been improved but there are some remaining issues that need to be addressed, as outlined below.Reviewer #3 (Recommendations for the authors):In general, the authors produced a stronger manuscript that addresses most of the concerns that were raised. In particular, the authors improved the overall clarity of their work both conceptually and methodologically, and they provide better data visualization. Besides, the scope and contribution of the paper is now communicated more clearly.Yet, a few aspects of the manuscript remain a bit obscure to me and I believe they should be further improved or at least addressed.First, after reading the paper, the pattern of results obtained from the combined EEG – bNIRS analysis does not seem as clear as I would have expected from the authors predictions. In figures 3-1 and 3-2, although some bNIRS channels (11 and 14) are overall more consistently reported than others as significant, as emphasized in the results and discussion, there is still quite some variability in the pattern of bNIRS to EEG channels coupling across chromophores (and frequency bands and conditions). How do the authors explain this variability? Shouldn't the different chromophore measurements reflect a similar phenomenon, and thus display similar coupling patterns with EEG channels? Is this the expected level of variability for this kind of dataset, or for this age range, population size etc….? It would be really informative if the authors could elaborate more on the reasons why we observe this variability in the discussion, and if possible provide recommendations on how to reduce it in further studies.

We thank the reviewer for their comment. We note that the reviewer also made a comment for us to clarify the FDR correction procedure and upon checking this (in order to include the exact details in the manuscript), we noted that the results figures that we had included after the previous revisions were not the FDR-corrected ones. The correct ones have now been included in Figure 3 supplement 1 and supplement 2 and are now much clearer. We see a more consistent, stimulus-dependent, localised pattern of associations between bNIRS channels and EEG electrodes. There is still a small amount of variation across chromophores, and we have added the following into the discussion on lines 340 – 395 which we hope will clarify why there is variability across chromophores and frequency bands:

“The strongest coupling was observed between temporo-parietal bNIRS channel 14 with parietal EEG locations Pz and PO4 for the social condition (for β and γ frequency bands). Meanwhile, for the non-social condition, coupling was observed between temporo-parietal bNIRS channel 14 with occipital and parietal EEG locations Oz, O2, PO8 and P10 (for theta and β frequency bands). While an overall consistent pattern of associations across chromophores and conditions was observed, some variability was also seen, particularly across frequency bands. This was expected and in line with previous EEG-fMRI studies that have demonstrated task-dependent variation in coupling between neural and haemodynamic activity across frequency bands [1]–[7]. For example, for resting state simultaneous fMRI and EEG, stronger coupling between the BOLD response and neural activity has been observed for the α band [8]. Meanwhile, for cognitive tasks, stronger coupling has been observed in the γ frequency band [9]. Scheeringa et al. (2011) investigated trial-by-trial coupling of EEG and BOLD activity and found that low- and high-frequency bands independently contribute to explaining BOLD variance. We therefore expected the frequency band showing the strongest coupling between bNIRS and EEG for each of the stimuli to vary. Further, while we did expect and observe significant overlap in associations between chromophores within each frequency band, some variability was seen. For example, for the social condition, no associations were observed in the low-frequency bands for any of the chromophores. Moreover, in the β frequency bands, all chromophores displayed significant associations between bNIRS channel 14 and Pz for the social condition and both HHb and oxCCO displayed significant associations between bNIRS channel 14 and O2, PO8 and C2. Similarly, in the γ frequency bands, both HbO_2_ and oxCCO displayed significant associations between bNIRS channel 14 and PO4. The variability that was observed between chromophores was limited mostly to the non-social condition. For example, only oxCCO displayed significant associations between bNIRS and EEG for the low-frequency theta and α frequency bands. It is well known that various components involved in neurovascular coupling undergo development postnatally, see the review by [18] for a full discussion. Briefly, there is extensive structural change within cerebral microvasculature including growth, extension and proliferation of new blood vessels [19], [20]. Further, studies have also demonstrated gradual development of vascular reactivity (i.e., change in vascular tone, vasoconstriction and vasodilation) [20], [21] which is necessary for the propagation of the NVC response [23]. Lastly, pericytes and astrocytes which are key components of NVC are also known to undergo development in size, number, connectivity and branching [24]–[27]. From the metabolic perspective, infant positron emission tomography (PET) studies demonstrate regional-specific, progressive increase in the cerebral metabolic rate of oxygen consumption (CMRO_2_) [14] while others evidence a developmental maturational change in oxidative metabolism [15]. In adults, previous research has also suggested that oxygen consumption is more spatially localised in comparison to changes in cerebral blood flow [10] and that oxCCO has distinct spatial distributions in the brain [11], [12], [13], indicating that energy metabolism may be more spatially specific. The spatial distribution of oxCCO in different brain regions currently remains unmapped in the developing infant brain, however. Therefore, taken together, given that there are extensive changes in cerebral vasculature as well as the metabolic environment and potential variability in the spatial distribution of oxCCO, it is expected that there will be some variability observed in the associations between the haemodynamics and metabolism with neural activity. In our study, we observed more consistent oxCCO – EEG associations across frequency bands and stimuli with more localised (fewer long-range) associations. Further studies with a larger sample size and longitudinal follow up can provide a clearer view on how NVC develops in infancy which will help explain some of the observed variability. Moreover, future studies with high density bNIRS arrays will provide clarification on the spatial distribution of oxCCO in the infant brain.”

Second, about the authors' predictions in the introduction, they mention that they expect to observe "coordinated increases in hemodynamics/metabolism and neural oscillatory activity" in the social condition, and "coordinated decreases in hemodynamics/metabolism and neural oscillatory activity" in the non-social condition. This wording suggests that this is a hypothesis that the authors will be able to test. However, the authors do not test this hypothesis further in the manuscript and it seems to me that the directionality of coordinated activity cannot be directly assessed using this GLM approach. I would therefore suggest to slightly reframe the phrasing of this paragraph to avoid that confusion.

The wording in the introduction and the discussion has been changed to the following and instead of the word coordinated, we have used the word simultaneous.

“We predicted that for the individual comparison of the social condition, we would observe positive associations between bNIRS and EEG measures, i.e. a simultaneous increase in haemodynamics/metabolism and neural oscillatory activity in the β and γ frequency bands (based on previous combined EEG – fMRI studies [1]–[7]) which would be localised to core social brain regions. We hypothesised that for the non-social condition, over the same brain regions, positive associations would be observed between bNIRS and EEG measures, but they would be a simultaneous decrease in haemodynamics/metabolism and oscillatory activity. We also expected simultaneous increases in haemodynamics/metabolism and oscillatory activity to be localised to the parietal brain region.”